# Rehabilitation of Individuals with Special Educational Needs through Music: An Accreditation Model Proposal [note 1]

**DOI:** 10.3390/bs14040329

**Published:** 2024-04-15

**Authors:** Hazan Kurtaslan, Ü. Ezgi Güleken

**Affiliations:** Department of Music, Faculty of Fine Arts, Akdeniz University, Antalya 07030, Türkiye; ezgiguleken@akdeniz.edu.tr

**Keywords:** rhythm, sing, dancing to music, grounded theory

## Abstract

This study aims to eliminate the current deficiency in the use of music in the field of special education, to increase musical activities in special education, and to present an accreditation model proposal to increase the skills of institutions, educators, and students. The research was designed with a grounded theory study pattern, one of the qualitative research methods. Data for the research were collected examining the necessary documents, and through semi-structured interviews with experts in both the field of special education and music. As a result of the interviews, it was concluded that music was used as both a goal and a tool in the education of individuals receiving special education and that different skills were developed through rhythm, melody, and lyrics. It has been revealed that individuals’ body coordination skills are improved through the rhythm in music, self-care skills are improved through melodies and lyrics, and behavioral disorders are corrected through rhythm, melodies, and lyrics. All these results led to the emergence of an accreditation model proposal to develop and rehabilitate individuals receiving special education through music.

## 1. Introduction

Accreditation means equivalence. Accreditation, which is also referred to as co-authorization, can also be said to be a system developed to ensure what is offered at certain standards. According to Aktan and Gencel [1], it is the determination of the competence of an institution or organization that is a candidate to produce a certain good or provide a service within the framework of determined standards. According to Ard, Beasley, and Nunn-Elison [2] (p. 80), accreditation is carried out by independent or third-party institutions voluntarily for institutions that want to participate in the practice, improve academic programs, ensure quality assurance of academic institutions and programs, and ensure quality improvements in educational quality. It can be expressed as the process of determining, implementing, and auditing standards. Özkale [3]; defines accreditation as continuous improvement of quality. Based on these definitions, “being accredited” according to Özkale means receiving a formal authorization statement about the quality of a department/program by an authorized institution as a result of periodic academic evaluations by accepted quality standards.

There are examples of accreditation in the private sector, education, healthcare, and engineering fields. There are many national quality agencies such as Engineering Education Programs Evaluation and Accreditation Association (MÜDEK), Science and Literature, Language and History-Geography Faculties Curriculum Evaluation and Accreditation Association (FEDEK), Education Faculties Programs Evaluation and Accreditation Association (EPDAD), VEDEK, TEPDAK, TPD, MİAK, and HEPDAK [4] (p. 59). These accreditation agencies have the right to grant registration period and registration authority at certain intervals.

Authorization of national accreditation bodies is carried out by the Higher Education Quality Council (YÖKAK). YÖKAK has three basic duties: It carries out external evaluation of higher education institutions, carries out the authorization and recognition processes of accreditation bodies, and finally ensures the internalization and popularization of quality assurance culture in higher education institutions. As of June 2023, the number of national accreditation institutions in Turkey with ongoing registration by YÖKAK is 30 and the number of organizations is 22 [5]. According to Doğan [6] (p. 503), accreditation in education in Turkey is a concept that was first updated in 1996 by YÖK and relevant units of YÖK for the restructuring of education faculties and equivalent institutions that train teachers, taking American and British practices as examples. With this method, which brings a new order and a new perspective to teacher training, continuous inspections ensure that institutions do not fall below a certain quality. In this context, Sakarya University Faculty of Education, Turkish Language Teaching Department became the first Turkish Department in Turkey to be accredited by EPDAD.

EPDAD is one of the organizations registered by YÖKAK. The EPDAD teacher education accreditation system is designed on three types of standards. These are initial standards, process standards, and product standards, respectively [7] In the accreditation list of undergraduate programs accredited by EPDAD, there are accredited departments affiliated with the education faculties of universities in Turkey.

When looked at in the context of the research topic, there are 10 special education teaching and 4 music teaching departments among the undergraduate departments accredited by EPDAD. Music teachers and special education teachers, who started their careers by graduating from various universities in Turkey, continue their duties in both public schools and rehabilitation centers to provide education to receiving special education. Graduates of special education teaching and music teaching educate and rehabilitate individuals receiving special education through music as well as field lessons in the private educational institutions where they are employed. Through music, observe that all people in need of special education improve and rehabilitate their disabilities. Music helps these individuals to develop language and social communication skills [8,9,10] psychomotor skills [11,12], emotional skills [13,14], physical skills [15,16,17]; and self-care skills [18,19]. Activities carried out with music, which is a tool for the development of individuals receiving special education, affect these individuals positively in cognitive, emotional [20], psychomotor, physical, mental, language, psychological, and behavioral aspects through the materials used in the activities, as well as plays an essential role in increasing their self-care skills and quality of life and in revealing their potential.

In this study, it was revealed that the music used in the development of individuals receiving special education was given by both special education teachers and music teachers, but both field teachers could not create a common program due to differences in individuals and the teachers followed their path. As a result of the research conducted in Turkey, we have observed that there is no program accreditation study for rehabilitation studies implemented through music for individuals receiving special education. This study aims to present an accreditation model that special education teachers and music teachers can apply for the rehabilitation of receiving special education in the special education institutions where they work. Based on the opinions of the teachers, an accreditation model has been suggested to rehabilitate receiving special education through music. An accreditation model proposal for the rehabilitation of individuals receiving special education through music will be gained for the first time with this study. The basic question determined for the research is as follows: What kind of theory or model can explain the establishment of standards for the rehabilitation of individuals receiving special education through music? An accreditation model proposal was developed based on this question. It is thought that this developed model will shed light on the change processes that will occur in the future. According to the purpose of this study, teachers’ opinions were consulted.

Answers to the following questions were sought for the research.

What are the types of activities used to improve the skills of individuals receiving special education to rehabilitate them through music?What are the materials used in activities to improve the skills of individuals receiving special education to rehabilitate them through music?What are the names of activities used to improve the skills of individuals receiving special education to rehabilitate them through music?

## 2. Materials and Methods

### 2.1. Research Design

The research was conducted with qualitative research methods. Qualitative research is in line with human nature and holistically examines, understands, and interprets events and phenomena in their natural environment. [17]. The grounded theory design was used in this study. In the literature, theory formation is also known as grounded theory. According to Glaser and Strauss [7], grounded theory is a research method in which data-based theoretical models or propositions are developed or contribute to existing theories as a result of examining behaviors, activities, and processes [21] (p. 32). Glaser and Strauss [7] define grounded theory as a process based on reason, analytical processes and unbiased observation, emphasizing induction, using qualitative and quantitative methods, and with four stages of coding. According to Ilgar and Ilgar [22], grounded theory has two popular approaches. The systematic approach of Strauss and Corbin [23] and the structuralist approach of Charmaz. In the more systematic and analytical method of Strauss and Corbin, researchers systematically search for a theory that explains the process, events or interaction on the topic. Charmaz presents a theory based on flexible studies and observations of researchers with a constructivist perspective, emphasizing the complexity of events, observations and a particular world and various local worlds, with multiple relationships [23]. One of these is the systematic approach of Strauss and Corbin [23]. In Strauss and Corbin’s systematic and analytical method, researchers systematically search for a theory that explains the process, events, or interaction on the subject. Strauss and Corbin’s systematic approach was adopted for the research. This study is based on obtaining the theory through data, rather than reconsidering a predetermined theory or developing that theory. The theory that would explain the resulting interaction regarding the process or events or the scope of the subject was systematically investigated. It is aimed to reveal the information embedded in the study data through grounded theory. In addition, the research was conducted based on Strauss and Corbin’s 3-stage coding: open coding, axial coding, and selective coding.

Individuals receiving special education are rehabilitated through music in schools, based on teachers’ programs. It is noteworthy that the process of rehabilitating individuals receiving special education through music works in different ways in schools and that there are no certain standards for these schools. To rehabilitate individuals receiving special education through music within the framework of certain standards, an accreditation model was developed using the opinions of teachers therefore, the grounded theory pattern was used in the research [24]. The basic question determined for the research is as follows: What kind of theory or model can explain the establishment of standards for the rehabilitation of individuals receiving special education through music? An accreditation model proposal was developed based on that question. According to the purpose of this study, teachers’ opinions were consulted.

### 2.2. Study Group

In the research, the study group was determined using the criterion sampling technique, one of the purposeful sampling methods. The study group of the research consists of special education teachers and music teachers who carry out music rehabilitation studies in public special education schools in Antalya. Important criteria for the research are that teachers have worked in special education schools for at least 2 years and have worked on rehabilitating individuals through music. Teachers work in the Muratpaşa and Konyaaltı districts, which are the central districts of Antalya province.

The interviews, including the pilot interviews, were conducted with a total of 6 teachers in the special education (3) and music branches (3) from 5 special education schools. The interviews held at the school and information about the instructors are shown in Table 1.

### 2.3. Data Collection Tools

Data were collected through documents and interviews appropriate to the scope of the research. Interviews and observations are known to be the most preferred data collection tools in grounded theory (theory building) studies. Document review is also used when deemed necessary for the purpose of this study [23] (p. 76). Therefore, documents and interviews were used as data collection tools in this study.

Individual interviews were held with special education teachers and music teachers working in special education schools. In addition, other data of this study were created with the curriculum obtained from schools, the documents and materials used, and the information regarding the subject of this study obtained from websites.

### 2.4. Data Collection

A qualitative data collection process was followed in the research. Documents were collected for the research and individual interviews were conducted voluntarily. According to many studies [25] (p. 149), in grounded theory approaches, data are generally collected using interviews, observations, diaries, and other written documents, or a combination of some other methods. In this study, data were collected based on documents and interviews.

In grounded theory studies, the interview is often the preferred type of data collection because it allows in-depth information to be obtained. Semi-structured interviews were conducted in the research [26]. For this purpose, a semi-structured interview form was prepared. In the semi-structured interview form, the researcher prepares the main questions he wants to ask in the interview text in advance and can direct the flow of the interview with additional questions based on what the participant said during the interview. This will prompt the participant to go into more detail in some of the answers. The interview and the analysis of the data obtained from the interview are analyzed simultaneously.

A consent form was prepared before the interview and signed by the participants. Information regarding the purpose of this study, the confidentiality of the participants’ identity information, and that the data will be used only for scientific purposes were clearly stated in the consent form, and signatures of the participant and the researcher were obtained. Interviews held between 18 May 2023 and 28 May 2023, with the decision of Akdeniz University Social and Human Sciences Scientific Research and Publication Ethics Board’s 280th Ethics Committee, dated 18 May 2023. During the interview, after the participant was informed about the research, the questions were asked. While preparing the interview questions, the relevant literature was scanned. Expert opinions were received for the questions that were thought to best reflect the subject and necessary corrections were made. Additionally, at the end of the pilot interview, the questions were reviewed and finalized. The interviews were recorded with a voice recorder with the permission of the participants. After each interview, the interview was transcribed and the coding was completed before starting a new interview. The transcripts were intended to be sent to the participants for participant confirmation, but were not sent after the participants stated that they did not need them. The transcripts and coding obtained from the interviews were re-evaluated by relevant field experts for their compatibility with the interview records. As a result of the evaluation, confirmation was received from expert opinions that the transcripts and coding were performed correctly. Before starting a new interview, a code list is obtained from the last interview and this code list is constantly compared throughout all interviews, and by following an inductive approach, if any deficiencies are detected, the codes in the transcript are added to the new code list or changes are made to the existing codes. Glaser and Strauss [8] named this form of analysis as constant comparative analysis. A total of 38 pages of raw data text were obtained as a result of the transcript for the research. The total interview time with all participants was 3 h and 48 min. The transcripts were intended to be sent to the participants after obtaining participant confirmation, but they were not sent as the participants stated that they did not need them. Additionally, participants were coded as A-B-C-D-E-F-G within the research.

### 2.5. Data Analysis

The research data were compiled through content analysis from qualitative data analysis. “Content analysis actually includes theme and stylistic analysis. In other words, theme analysis and stylistic analysis are needed in order to conduct content analysis, because content analysis is a deeper, more comprehensive and complex form of theme and stylistic analysis. The main purpose of content analysis is to interpret the data, which we thematize in the first stage and make sense of in the second stage, comparatively in the last stage by including the researcher’s own interpretations” [27]. Content analysis also includes the theme and descriptive analysis [28]. The interviews were converted into transcripts by the researcher, and the accuracy of the transcripts was examined and confirmed by obtaining expert opinions. Interview transcripts were analyzed and codes were made. Constant comparative analysis was also used in this study, as it was a process of explaining the concepts, themes, and their relationships with each other, developed in line with the data obtained. According to Hancock [29], constant comparative analysis means conducting the data collection and data analysis process together and this is often used in grounded theory studies. Glasser and Strauss called this process “constant comparative analysis”. In this process, data are analyzed immediately after collection, and the resulting concepts, facts, and processes are later included in the data collection stages. In such a process, the interview form as a data collection tool is in a semi-structured state at the first stage of data collection and does not take its final form until the end of the data collection process. In fact, in some studies, there may be significant differences between the interview questions discussed at the beginning and the questions created towards the end of data collection [27] (p. 76). The resulting concepts and themes provide a meaningful explanation of the focus of the research. This explanation is the theory that emerges based on data [27] (p. 77). The data collected in this study were analyzed simultaneously. The analysis process was carried out with the Nvivo10 program.

In the research, the transcripts obtained from the interviews were analyzed and coded. There are three types of coding stages in grounded theory studies: open coding, and selective coding.

Data analysis of this study started with an open coding process. According to Strauss and Corbin [23] (p. 12), open coding is an interpretive process in which data are analytically divided into parts. The purpose of open coding is to gain a new insight for the researcher in thinking and interpreting the phenomena reflecting the data by separating the data according to certain standards [23] (p. 153). According to Strauss and Corbin [23] (p. 72–73) the researcher can code the data line by line, code over this or paragraphs, or code over the entire data collected. In the open coding process of this study, the researcher first read the data line by line and coded it by underlining the words, names, and sentences that were thought to be related to the research question and gave meaning to each line representing the concepts. Then, both sentence and paragraph analyses were applied. The emerging concepts are categorized. The names of the categories were created from codes inspired by the literature and from the words used by the participants themselves.

After open coding is completed, the axial coding phase begins. According to Işık [30], in the axial coding phase, comprehensive supercategories can be obtained by determining the connections between the categories and concepts obtained in the open coding phase and associating them. After open coding was performed in this study, a we determined the relationship between categories or subcategories [31].

In the open and axis coding stages, the opinions of two experts who would contribute with their knowledge and experience in the subject of the research and the method of the research were consulted. The data obtained for the research were coded independently by two experts and the consistency between the coders was checked. Kappa statistic (analysis) was used to find the agreement between the data coders, also known as independent observers. Kappa statistic (κ), which is frequently used in determining inter-rater reliability, was proposed by Cohen [32]. It was developed to determine the degree of agreement between two raters scoring at the classification level [8]. The κ statistic, which was limited to two raters, was generalized by Fleiss [33] so that it could be used to determine the agreement between more than two raters [13]. The range of agreement values in kappa analysis; 01–20 shows “poor agreement”, 21–40 shows “below average agreement”, 41–60 shows “moderate agreement”, 61–80 shows “good agreement” and 81–0.99 shows “very good agreement” [34] (p. 362). The kappa value in this study was 858 for music teachers and 894 for special education teachers. These values can be interpreted as “very good agreement” according to Cohen kappa intervals.

Finally, selective coding was performed in the research. Selective coding is the stage of combining categories determined by axial coding. According to Strauss and Corbin [23] (p. 116–117), it is the process of selecting the core category, systematically associating it with other categories, verifying these relationships, and filling in the categories that need further purification and development). During the selective coding stage, a core category (a core concept) was determined for the research and it was associated with other categories (other concepts). All categories are united around the core category.

The core category in the research was named “rehabilitation of individuals receiving special education through music”. At the end of the research process, the theory was verified by researching and examining the literature on the subject.

### 2.6. Validity and Reliability

In qualitative research, the concept of validity (internal and external validity) reveals the accuracy of the findings in the scientific study, and the concept of reliability (internal and external reliability) reveals the repeatability of the findings.

### 2.7. Internal Validity (Credibility)

In qualitative research, the concept of internal validity, i.e., the concept of plausibility, reveals the extent to which the findings accurately describe reality. To ensure the internal validity of the research, it is important to diversify the data. By using more than one data collection method in the research, the internal validity of the research is increased. In the research, data diversity was ensured by examining documents, scanning the literature, and conducting interviews. In order to ensure credibility in the study of Ersoy [23] (p. 157), “help was obtained from impartial experts who were independently involved in the processes including research design, preparation of interview questions, analysis of the data obtained and who had knowledge and experience on the subject of the research and qualitative research method”. In creating the research design conducted and developing the interview form according to the data collection technique, a framework for the research topic was created and expert opinion was obtained after scanning the relevant literature. Mutual trust was established by written documents and signatures stating that the information provided by the participants in the interviews would be used only for scientific purposes and that their identities would be kept confidential. Transcripts of the interviews were sent to the participants and their confirmation was obtained. Thus, it was revealed that the collected data reflected the real situation for the research. After collecting the data, during the analysis process, the themes, categories, and sub-themes revealed in the content analysis, the relationship between each other, and the relationship between each theme were checked, and then expert opinions were obtained and confirmation was provided.

### 2.8. External Validity (Transferability)

In qualitative research, the concept of external validity, or transferability, refers to the ability to transfer research findings and research methods accurately and intelligibly to similar topics, situations, settings or groups (participants). The ability to transfer research findings and methods from one group to another, or the extent to which the findings of a particular study can be applied in other contexts or on other subjects/participants, is referred to as transferability [35] (p. 1290).

In order to ensure the external validity of the research, subheadings such as the research design, study group, sampling type, data collection tools, data collection, data analysis, analytical generalization, and comparisons with other studies on the subject are covered in detail in this study. In the information regarding the findings, themes, categories, and sub-themes were determined in line with the data obtained from the participants, and the opinions were described with direct quotations.

### 2.9. Internal Reliability (Confirmability)

In qualitative research, the concept of internal reliability, i.e., consistency, reveals that the topic under investigation is analyzed in the same way over time and it is important that it is consistent and stable across researchers [36] (p. 90).

İnternal reliability includes evaluating the objectivity of the collected data and whether the findings are supported by data. Accordingly, to ensure the internal reliability of the research, all data in the findings section are presented without comment. The opinions received from the participants are presented in the findings section with direct quotes [37]. The findings of the research were revealed not based on the researcher’s judgments, but based on the answers from the participants’ opinions. In addition, the analysis process of the data based on the opinions obtained from the participants was continued by obtaining expert opinions, independent of the researcher. The data obtained from the interviews were coded by the researcher. They were also coded separately by an experienced field expert. A third expert re-coded the data and compared them, thus demonstrating the consistency between the evaluators in this study. Kappa value, which is the calculation of the consistency ratio between the evaluators, was calculated and thus the agreement between the evaluators in the research and the reliability of the coding were examined.

### 2.10. External Reliability (Consistency)

External reliability refers to the extent to which the findings of this study are consistent and can be replicated. External reliability is critical because it assesses the methodological rigor of this study and adherence to a systematic research process in how data are collected and analyzed [38].

It is important to convey the information obtained directly from the data in the findings section of the research without being affected by the researcher’s emotions. Adherence to research processes is assessed in the way data are collected and analyzed. Accordingly, all the data obtained by the researcher, data collection tools, data collection process, the codes used in the analysis of the data, and the analysis of the data were submitted to an expert review and evaluation and were examined for consistency.

### 2.11. Findings

In this section, answers were sought to three research questions. The headings obtained based on the research questions also constitute the main headings of the research model. These headings are as follows: “Types of Activities Used through Music”, “Materials Used Through Music”, and “Names of Activities Used Through Music” in the rehabilitation process of individuals receiving special education. “Types of Activities Used to Develop Skills”, “Materials Used to Develop Skills”, and “Descriptions of Activities Used to Develop Skills”. These main headings also represent the sub-models of this study.

### 2.12. Findings Regarding the First Research Question

According to Table 2; When we look at the types of activities that teachers working in special education schools use to develop the skills of individuals receiving special education through music, it is seen that teachers mostly prefer the types of activities such as listening to music with/without lyrics and dancing to music, and that these themes develop emotional, physical and social development skills. It is also seen that they prefer activities such as singing, keeping rhythm, and singing and playing in a group, and that these themes develop language, self-care, motor, physical and social skills.

According to Figure 1; sample opinions of the participants regarding the type of activity they prefer most, which is listening to music with/without lyrics and dancing to music, are given below:

“A child who never communicates and never makes eye contact can communicate when I make him/her listen to music consisting of melodic sounds.”(A1,3)

“I make children listen to music based on the types of music they like, that is, the music they respond to. Sometimes it is classical music, sometimes traditional dance music or a folk song.”(B1,3)

“I do not agree with teachers who think that listening to music is not beneficial. Music calms the child down, I play a song he likes without him knowing. This music definitely leaves an impact on the child’s soul.”(D1,3)

“Let me give an example from the activities, for example, there is a game called 7-step dance. In this game, there is dance, at first there are movements with a certain rhythm and there is music, the children do these movements one by one, accompanied by a group.”(A1,4)

“We use dance in group play. We use their dancing to do activities together and socialize. I strive for the child’s physical development and mobility through music and dance. It also improves children in physical aspect.”(B1,4)

**Figure 1 behavsci-14-00329-f001:**
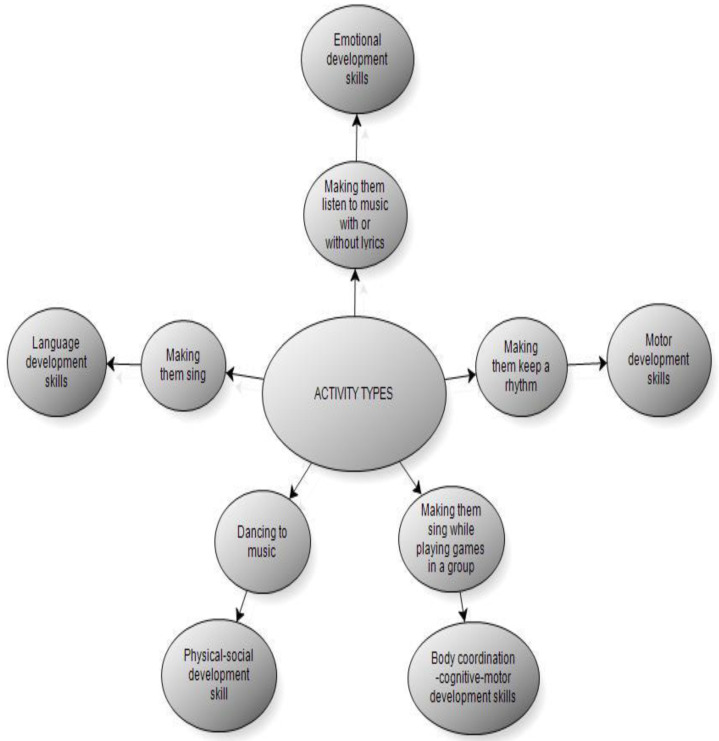
Types of Activity Used to Improve the Skills of Individuals Receiving Special Education to Rehabilitate them through Music (Sub-Model).

### 2.13. Findings Regarding the Second Research Question

According to Table 3, When we look at the materials used by teachers working in special education schools in activities to improve the skills of individuals receiving special education through music, it is seen that the teachers prefer maracas and drums first and that these instruments improve body coordination, cognitive and motor development skills; and it is also seen that language, body coordination, cognitive and motor development skills develop with the preference of songs and lyrics in the second place. Finally, it is seen that teachers use Orff instruments, flute, baglama, drum set, qanun, cymbal, oud, and piano. According to Figure 2, it is seen that these instruments improve body coordination, and cognitive and motor development skills in individuals, too. According to this:

Sample opinions of the participants regarding maracas and drums, which are the materials they use most in the activities, are given below:

“……we usually start with the simplest musical instruments. For example, we start with a drum and a maraca. Using more than one instrument at the same time or doing one task while doing the other affects the improvement of divided attention and cross-brain development…”(A2,2)

“……I had another student with cerebral palsy. I was making him practice rhythm with maracas, which was an activity that improved his body-hand-eye coordination and gross motor skills.”(B2,2)

**Table 3 behavsci-14-00329-t003:** Table on Materials Used to Improve Skills for Rehabilitating Individuals Receiving Special Education through Music.

Category	Theme	A	B	C	D	E	F
Language Development Skills	Songs and lyrics		√		√		
Body Coordination, Cognitive and Motor Development Skills	Orff Instruments	√					
	Flute				√		
	Bağlama				√		
	Melodica		√				
	Maracas	√	√	√	√		
	Drum	√		√	√		√
	Percussion				√		
	Kanun				√		
	Cymbal			√			
	Ud					√	
	Piano	√					

**Figure 2 behavsci-14-00329-f002:**
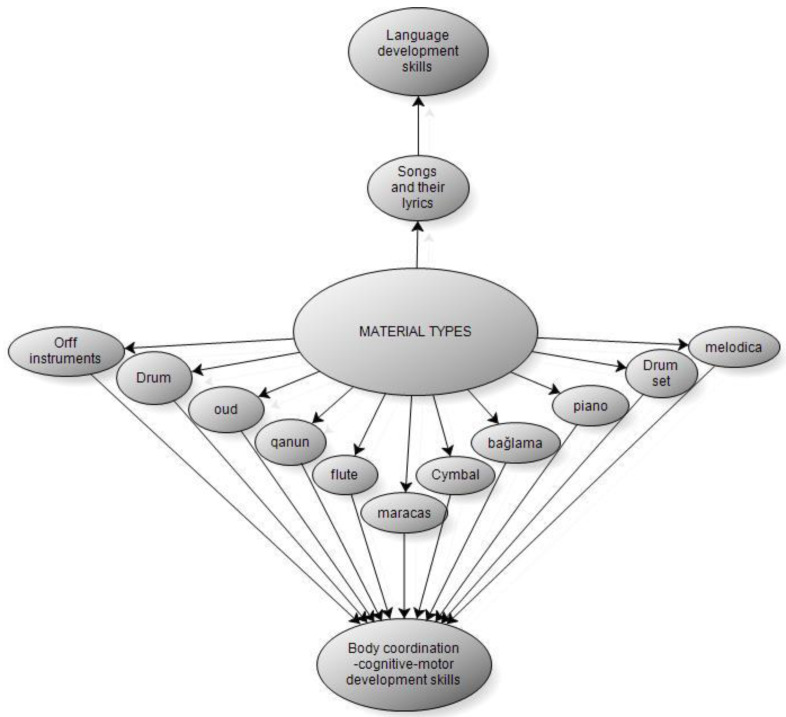
Materials Used in Activities to Improve the Skills of Individuals Receiving Special Education to Rehabilitate them through Music (Sub-Model).

### 2.14. Findings Regarding the Third Research Question

According to Table 4, when we look at the names of the activities, under the theme of allowing them to sing, used to improve the skills of individuals with special education to rehabilitate them through music by teachers working in special education schools, it was observed that they sing the songs such as “Let’s go pee, mom”, “Chubby Cheeks”, “Old MacDonald Had a Farm”, “Pee-pee on the Potty”, “A Little Lion”, “Walnut Man”, “Red Fish”, “Dinosaur”, “Sculpture”, “Long Live Our School” and “Morning Sport” and that these songs improved the language and self-care skills of individuals. It was observed that teachers mostly use classical music, followed by popular music and traditional dance music from Ankara, under the theme of listening to music with/without lyrics. Apart from this, songs from Hadise, Gazan, Luka, and Pepe are used. It is seen that the music used improves the emotional development skills of individuals. It is seen that teachers mostly prefer traditional dance music from Ankara under the theme of dancing with music. It is seen that this music also improves the physical and social development skills of individuals. Sample opinions of the participants regarding the names of the songs used in the singing activity are given below:

“I will give an example of a song for self-care skills. Let’s say potty training, for example. There is a song, ‘Let’s Go Pee, Mom’, which has lyrics like ‘hold it when pee comes, hold it when poop comes’. I also use the songs ‘Red Fish’ and ‘Walnut Man’ a lot in our activities.”(A3,1)

“There are songs about cleaning the body, hands, and nails. For example, there are lyrics like ‘my hands are chubby, when they get dirty it is funny’. We motivate them to clean by getting support from children’s songs. I use the song ‘Pee-pee on the Potty’ for self-care, and I also try to use the songs ‘Old MacDonald’s Farm’ and ‘A Little Lion’ in our activities.”(B3,1)

“Children love ‘Dinosaur’ song very much. We do an activity with “Sculpture” song as an equivalent to gym class, and there is also a music called ‘Morning Exercise’, which we use for the children as well.”(C3,1)

**Table 4 behavsci-14-00329-t004:** Table of Activity Names Used to Improve Skills for the Rehabilitation of Individuals Receiving Special Education through Music.

Category	Theme	Sub Theme	A	B	C	D	E	F
Language Development Skills	Singing	Tuvaletim geldi anne	√					
		Ellerim tombik		√				
		Ali babanın çiftliği		√				
		Güle güle bezlere		√				
		Bir küçücük aslancık		√				
		Ceviz adam	√					
		Kırmızı balık	√					
		Dinazor			√			
		Yaşasın Okulumuz				√		
		Heykel			√			
		Sabah sporu			√			
Emotional Skills	Listening music with/without lyrics	Klasik müzikleri	√	√	√	√		√
		Popüler Müzikler		√	√	√		√
		Hadise şarkıları				√		
		Sevdan Olmasa				√		
		Gazan						√
		Luka						√
		Ankara havaları				√	√	√
		Pepe’s songs		√				
Physical/Social Skills	Dancing to music	Ankara havaları				√	√	√

According to the Figure 3 and Figure 4, emerging from the research, in the rehabilitation of individuals receiving special education through music, the type of activity the teachers prefer, the materials used in the activities, and the names of the activities constitute the basic framework. Within the framework of this framework, rehabilitation of individuals through music appears to affect emotional, language, body, physical, social, and motor development skills.

## 3. Results

According to the first result obtained from the research findings, when the types of activities used by teachers to improve the skills of individuals receiving special education to rehabilitate them through music are examined, it is seen that all of the teachers make the individuals listen to music with or without lyrics and dance with music. It was revealed that these types of activities also improve emotional development skills and physical and social development skills.

According to the results of the research, when the materials used by teachers in activities to improve the skills of individuals receiving special education to rehabilitate them through music are examined, it is seen that the majority of teachers use maracas and drums. It was concluded that these materials improve physical coordination, and cognitive and motor development skills in individuals.

According to the results of another research, when the names of the activities used by teachers to improve the skills of individuals receiving special education to rehabilitate them through music were examined, it was seen that the names of the activities were grouped under four main headings. These are allowing them sing, listening to music with/without lyrics, and dancing to music. It was observed that teachers prefer different songs and improve their language development skills with songs. It was revealed that the pieces that teachers use under the theme of listening to music with/without lyrics improve the emotional development skills of individuals. It was observed that teachers use similar melodies in the theme of dancing with music, thus enabling individuals to develop physical and social development skills.

## 4. Discussion

Types of activities carried out to rehabilitate receiving special education through music contribute to their language, psychomotor, social, physical, mental, and psychological development, ability to express themselves, bond with their environment, sense of achievement, self-confidence and self-care [39]. In the module guidebook prepared by the Ministry of National Education for Special Education, it is stated that the activities carried out to rehabilitate receiving special education through music contribute to the self-care, motor, cognitive, social, emotional, and language development of individuals [40]. As music activity types, listening to and distinguishing sound/music, singing, keeping rhythm, dancing with creative movements, and musical stories were emphasized [41]. In this study conducted with the activity types in the module guidebook presented by Megep, it is seen that there are similarities in activity types such as making people sing, listening to music with/without lyrics, keeping rhythm, and dancing to music. In the study conducted by Akın [42], the effect of educational games on the development of basic motor skills of 60–72-month-old children was examined with an experimental method and in addition to musicals, activity types consisting of musical dances were designed by using the plays, and the basic motor skills and hand-arm coordination skills of children were developed [43]. With this study, as in Akın’s study, the development of individuals is ensured through playful dancing and singing with music in a group. Similarly, in the study conducted by Eren [44], it was observed that adolescents receiving special education needs developed both motor (small and large muscle groups) skills and social skills through singing, rhythmic games, and dance activities.

In the research conducted by Turan [45], the majority of teachers working in special education schools (75.5%) stated that in musical activities, it is not enough for the student to understand the lesson by just listening, but also that the use of materials contributes to the teacher’s performance, and the student’s attention will be attracted with the material. In the module guidebook prepared by Megep [40] (p. 58), for special education, with the Orff teaching method, it is recommended to use the body as a rhythm instrument and to use materials such as drums, tambourines, cymbals, rhythm sticks, triangles, maracas, metallophones and xylophones (www.megep.gov.tr, accessed on 29 March 2024). The fact that the Orff instruments mentioned in Megep’s material suggestions are also included in the material use in this research shows that both studies are similar in this aspect. In many studies, it was observed that children improved their hand-arm (body) coordination and motor skills (small and large muscle groups) thanks to rhythm instruments.

According to Turan, “the first step in the musical development of students receiving special education is singing with the student” [46] (p. 39). Singing in music class not only pleases the students but also increases their understanding and improves their language development skills [47,48].

With the research conducted, a model for accrediting a program to rehabilitate individuals in private educational institutions through music was presented. Whether it is a special education teacher or a music teacher working in a private education institution, it is important to benefit from the influence of music while providing that education in order to popularize and make effective education and training aimed at rehabilitating individuals. For this purpose, it is recommended to provide training and accreditation of institutions and organizations upon successful completion of the training.

## Figures and Tables

**Figure 3 behavsci-14-00329-f003:**
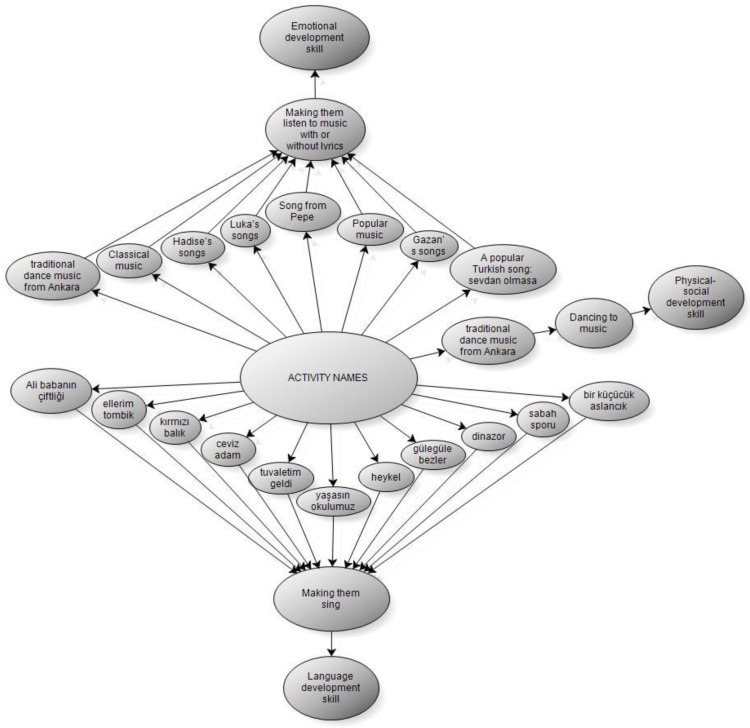
Names of Activities Used to Improve the Skills of Individuals Receiving Special Education to Rehabilitate them through Music (Sub-Model).

**Figure 4 behavsci-14-00329-f004:**
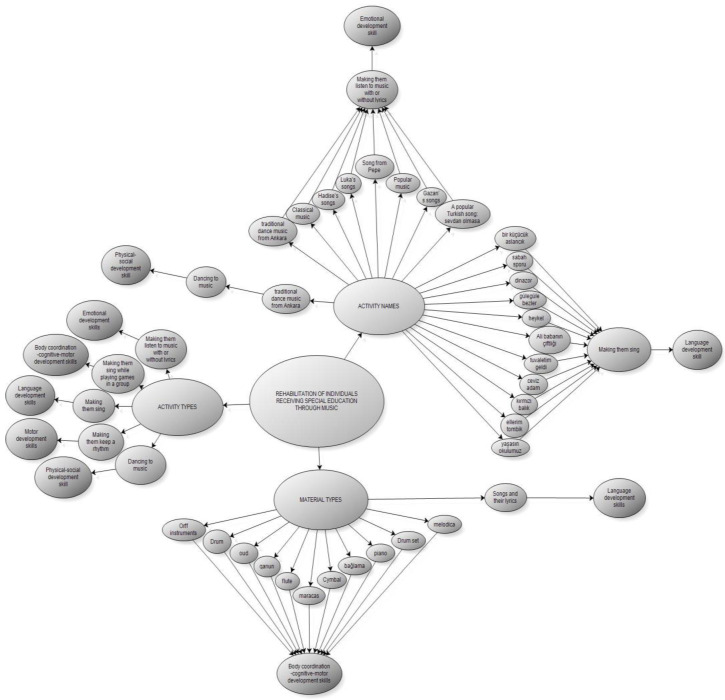
Model Developed for the Rehabilitation of Individuals Receiving Special Education through Music (Main Model).

**Table 1 behavsci-14-00329-t001:** Demographic Information of Participants.

Participants	Institution	Central District	Branch	Tenure of Office
A	Special education school	Muratpaşa	Special education teacher	5 years
B	Special education and practice school III Level	Muratpaşa	Special education teacher	10 years
C	Secondary school	Konyaaltı	Special education teacher	20 years
D	Special education secondary school and industrial school	Konyaaltı	Music teacher	23 years
E	Special education and practice school III Level	Muratpaşa	Music teacher	28 years
F	Special education and practice school III Level	Muratpaşa	Music teacher	17 years

**Table 2 behavsci-14-00329-t002:** Table on Types of Activities Used to Improve Skills for the Rehabilitation of Individuals Receiving Special Education through Music.

Category	Tema	A	B	C	D	E	F
Language Development Skills	Singing	√	√	√	√		√
Motor Development Skill	Rhythm	√	√	√	√		√
Emotional Development Skill	Listening to music with/without lyrics	√	√	√	√	√	√
Physical and Social Development Skill	Playful singing with group	√	√				√
	Dancing to music	√	√	√	√	√	√

## Data Availability

Research scholars interested in working with the data used in this study are asked to contact the corresponding author with a short description of the planned study.

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
