# Peer review of "Rehabilitation of Individuals with Special Educational Needs through Music: An Accreditation Model Proposal"

_behavsci, 2024, doi:10.3390/bs14040329_

Round 1

Reviewer 1 Report

Comments and Suggestions for Authors

Application of grounded theory qualitative research design and procedures is excellent. Criterion sampling is identified. Two sources of data, interviews and archival document analysis, are explained. Coding methods are specifically documented. Very well constructed visual inserts to supplement summary narrative of findings. I would add a few more recommendations for practical application and future related research. Overall, very well done study.

Comments on the Quality of English Language

In general the quality of writing is very good. Some APA 7 corrections remain (e.g., "semistructured" is now written as one word according to APA 7). Also, please check spelling of Barney Glaser's last name (one 's') in the Glaser and Strauss in-text citation and reference.

Author Response

Dear referee, thank you very much for the corrections you provided. we have made the corrections you indicated. We have corrected "semistructure" and "Glaser". thank you again. Please see attachement. Respectfully.

Reviewer 2 Report

Comments and Suggestions for Authors

Some areas to consider:

- Given the international scope of the journal, it may be important to explain why you are choosing the language "Special Needs." This language is outdated, and sometimes offensive, in some areas.

- In some of the data, the phrase "making them" is present when speaking about young people. This, too, is problematic. It will be important for the authors to speak to the agency of young people in these programs, and what is meant by "making them" do things.

Some slight typos/errors:

- line 26 has an additional period after "standards"

- Lie 47 - is it "Board" or "Council" (for YOKAK?)

- The graphs on line 11 and is difficult to read. This may be a formatting problem for the editors, not the authors. However, the authors should provide in-depth image descriptions for all graphs in order to make the article accessible.

Comments on the Quality of English Language

The English is great. There are a few errors, but nothing major.

Author Response

Dear referee, upon your request, we used the term "receiving special education"
instead of "special needs". We used the term "council" for YÖKAK.Instead of "making them", the terms "letting" and "allowing them" are used.
We made the graphics slightly larger and more readable.
We couldn't understand what you wanted after the word "standards" on line 26.
The corrected version of the article is attached. Regards.

Reviewer 3 Report

Comments and Suggestions for Authors

Qualitative research based on interviews with special education and music teachers. Data analysis was conducted using Nvivo.

It could be interesting to explain why there were differences in the interview questions created at the beginning and at the end, as well as provide some examples.

Grounded theory is explained in three stages, needing to be more specific in the research presented.

Methodology with well-argued validity and reliability, if not explained how it was done.

It would be necessary to specify the aspects to be rehabilitated with music, as they show activities included in the official curriculum and it is necessary to know the starting point to assess that the program is accredited for rehabilitation

Author Response

Dear referee, we have made your corrections. İnterview questions are compatible with each other. Grounded theory is explained more fully. Kappa analyzes were added. As a result of the resaarch, it has been observed that there is no program
accreditation study in TURKEY for rehabilitation studies carried out through
music for individuals with special needs.
Please, See the attached file for the corretctions made. Regards
